# Regulatory Roles of Caspase-11 Non-Canonical Inflammasome in Inflammatory Liver Diseases

**DOI:** 10.3390/ijms23094986

**Published:** 2022-04-30

**Authors:** Young-Su Yi

**Affiliations:** Department of Life Sciences, Kyonggi University, Suwon 16227, Korea; ysyi@kgu.ac.kr; Tel.: +82-31-249-9644

**Keywords:** inflammation, caspase-11, non-canonical inflammasome, NAFLD, NASH, liver injury

## Abstract

An inflammatory response consists of two consecutive steps: priming and triggering, to prepare and activate inflammatory responses, respectively. The cardinal feature of the triggering step is the activation of intracellular protein complexes called inflammasomes, which provide a platform for the activation of inflammatory signaling pathways. Despite many studies demonstrating the regulatory roles of canonical inflammasomes in inflammatory liver diseases, the roles of newly discovered non-canonical inflammasomes in inflammatory liver diseases are still largely unknown. Recent studies have reported the regulatory roles of the caspase-11 non-canonical inflammasome in inflammatory liver diseases, providing strong evidence that the caspase-11 non-canonical inflammasome may play key roles in the pathogenesis of inflammatory liver diseases. This review comprehensively discusses the emerging roles of the caspase-11 non-canonical inflammasome in the pathogenesis of inflammatory liver diseases, focusing on non-alcoholic fatty liver disease (NAFLD), non-alcoholic steatohepatitis (NASH), and inflammatory liver injuries and its underlying mechanisms. This review highlights the current knowledge on the regulatory roles of the caspase-11 non-canonical inflammasome in inflammatory liver diseases, providing new insights into the development of potential therapeutics to prevent and treat inflammatory liver diseases by targeting the caspase-11 non-canonical inflammasome.

## 1. Introduction

Although inflammation is a body-protective innate immune response, chronic inflammation is a key determinant of numerous inflammatory diseases and cancers [1,2]. An inflammatory response consists of two main steps, priming and triggering. Priming is a preparation step of inflammatory responses by upregulating the expression of inflammatory molecules while triggering is an activation step of inflammatory responses by activating inflammasomes, which are intracellular protein complexes providing the platforms of inflammatory signaling pathways [3,4]. Inflammasomes are categorized into canonical and non-canonical inflammasomes. The initially discovered canonical inflammasomes include nucleotide-binding and oligomerization domain (NOD)-like receptor (NLR) family inflammasomes (NLRP1, NLRP3, NLPC4, NLPR6, NLRP9, and NLRP12 inflammasomes) and non-NLR-family inflammasomes (absent in melanoma 2 (AIM2) and pyrin inflammasomes) [4,5]. Recently identified non-canonical inflammasomes include mouse caspase-11 and human caspase-4 and -5 non-canonical inflammasomes [6,7,8,9]. Although many studies have demonstrated the role of canonical inflammasomes in inflammatory responses and diseases [10,11], the regulatory roles of non-canonical inflammasomes, which were recently discovered in inflammatory responses and diseases, remain largely unknown.

Inflammation also induces liver diseases. Non-alcoholic *fatty liver* disease (*NAFLD*) is a chronic disease caused by excessive fat accumulation and inflammation in the liver. NAFLD can develop into non-alcoholic steatohepatitis (NASH), an aggressive form of fatty liver disease, which is characterized by liver inflammation and may progress to cirrhosis, liver injury, and liver failure [12,13]. Many studies have reported that canonical inflammasomes play critical roles in inflammatory liver diseases by promoting inflammation-induced injury in the liver [14,15,16]. Interestingly, recent studies have reported that non-canonical inflammasomes are also key players in inflammatory liver diseases and injury. This review summarizes and discusses the studies that highlight the regulatory roles of caspase-11 non-canonical inflammasomes in inflammatory liver diseases and injury, which can provide insight into the development of novel and potential therapeutics for inflammatory liver diseases by selectively targeting the caspase-11 non-canonical inflammasome.

## 2. The Caspase-11 Non-Canonical Inflammasome

### 2.1. Structure and Activation of the Caspase-11 Non-Canonical Inflammasome

The caspase-11 non-canonical inflammasome was first discovered in the 129S6 mouse strain, which has a polymorphism in the *caspase-11* gene locus, resulting in the expression of truncated and non-functional proteins [6]. The *caspase-11* gene is not found in humans; instead, *caspase-4* and *-5* genes have been identified as human homologs of the mouse *caspase-11* gene, and studies have demonstrated that the *caspase-4* and *-5* genes generate caspase-4/5 non-canonical inflammasomes in humans [9]. Unlike canonical inflammasomes, non-canonical inflammasomes have similar structures. Mouse caspase-11 and human caspase-4/5 consist of an N-terminal caspase recruitment domain (CARD), followed by two catalytic domains: a p20 large catalytic domain and a p10 small catalytic domain at the C-terminus (Figure 1A). Despite the same molecular architecture being among non-canonical inflammasomes, their sizes are different, and the amino acid lengths of mouse caspase-11 and human caspase-4/5 are 373, 377, and 434, respectively (Figure 1A).

Canonical inflammasomes are activated in response to their specific ligands [4,5]. However, lipopolysaccharide (LPS), an endotoxin derived from Gram-negative bacteria, has been identified as the only ligand that activates non-canonical inflammasomes [6,7,8,9]. Once LPS enters the host cells via receptor-mediated endocytosis [17], mouse caspase-11 and human caspase-4/5 sense intracellular LPS by direct binding [18,19,20,21]. The direct sensing of LPS by caspase-4/5/11 is mediated by the molecular interaction between LPS lipid A motifs and caspase CARDs to form LPS-casaspe4/5/11 complexes (Figure 1B). LPS-caspase-4/5/11 complexes, in turn, are oligomerized by direct CARD-CARD interaction, followed by the activation of non-canonical inflammasomes (Figure 1C) [18,19,20,21].

### 2.2. Caspase-11 Non-Canonical Inflammasome-Activated Signaling Pathways

As described earlier, the direct interaction between LPS and caspase-11 induces the oligomerization of the caspase-11 non-canonical inflammasome. The oligomerized caspase-11 non-canonical inflammasome is subsequently activated by self cleavage at the 285 aspartic acid residue (D285), and this enzymatic activity is mediated by the 254 cysteine residue (C254) of caspase-11 [22]. The two main inflammatory signaling pathways are activated by the caspase-11 non-canonical inflammasome. The activation of the caspase-11 non-canonical inflammasome induces the proteolytic cleavage of gasdermin D (GSDMD) at the 276 aspartic acid residue (D276), resulting in the generation of GSDMD N-terminal (N-GSDMD) and C-terminal fragments (C-GSDMD). N-GSDMD then moves to the cell membranes and generates GSDMD pores in the membranes, leading to inflammatory cell death, known as pyroptosis [18,19,20,21]. The activation of the caspase-11 non-canonical inflammasome also induces the proteolytic activation of caspase-1, and the active caspase-1 subsequently facilitates the proteolytic maturation and secretion of pro-inflammatory cytokines, interleukin (IL)-1β and IL-18, through the GSDMD pores, leading to the augmentation of inflammatory responses [18,19,20,21].

Although the caspase-11 non-canonical inflammasome induces the secretion of pro-inflammatory cytokines by activating caspase-1, it indirectly activates caspase-1 via functional cooperation with the NLRP3 canonical inflammasome. NLRP3 canonical inflammasome, the most-studied inflammasome, is activated in response to various pathogen-associated molecular patterns (PAMPs) and danger-associated molecular patterns (DAMPs). Among the PAMPs and DAMPs, the potassium ion (K^+^) efflux induced by GSDMD pore-mediated membrane damage and gate proteins, such as P2X7 channels, pannexin 1 channels, and bacterial pore-forming toxins, play a key role in the activation of the NLRP3 canonical inflammasome [4,5]. Recent studies have reported that the activation of the caspase-11 non-canonical inflammasome induces K^+^ efflux through gate proteins and GSDMD pore-mediated membrane damage, leading to the activation of the NLRP3 canonical inflammasome [23,24,25]. The activated NLRP3 canonical inflammasome then directly activates caspase-1, leading to the maturation and secretion of pro-inflammatory cytokines. The caspase-11 non-canonical inflammasome-activated caspase-1 mediated by the NLRP3 canonical inflammasome strongly suggests that caspase-11 non-canonical inflammasome-activated inflammatory responses are accomplished by functional interplay with the canonical inflammasome, rather than functioning in a canonical inflammasome-independent manner. The caspase-11 non-canonical inflammasome-activated inflammatory signaling pathways are described in Figure 2.

## 3. Regulatory Roles of the Caspase-11 Non-Canonical Inflammasome in Inflammatory Liver Diseases

### 3.1. NAFLD

NAFLD is the most prevalent chronic metabolic disease caused by the accumulation of fat in the liver, which affects a quarter of the global population and is likely observed in people who are overweight or obese [26]. NAFLD includes a wide range of fatty liver diseases, including fibrosis, cirrhosis, NASH, and hepatocellular carcinomas. NAFLD is associated with chronic inflammation in the liver, which causes systemic alterations in the immune system [27,28]. Obesity directly correlates with inflammatory responses and the accumulation of inflammatory cells, which contribute to chronic low-grade inflammation [29,30] and play a critical role in insulin resistance and NAFLD development [31]. In addition, a large number of innate immune cells that induce inflammatory responses, such as macrophages, monocytes, and neutrophils, are actively involved in the onset of chronic inflammation in the liver with NAFLD [32,33,34,35], indicating that NAFLD is a chronic inflammatory liver disease.

Given the evidence that NAFLD is caused by inflammation, studies have investigated the role of inflammasomes in NAFLD and demonstrated that the canonical inflammasome, particularly the NLRP3 inflammasome, plays a critical role in NAFLD pathogenesis [14,15,16,36,37,38]. Recent studies have also reported the regulatory role of non-canonical inflammasomes in NAFLD pathogenesis. Anderson et al. investigated the role of the caspase-11 non-canonical inflammasome in steatotic allograft-induced liver inflammation and injury. Steatotic allograft increased endoplasmic reticulum (ER) stress, which led to liver inflammation and injury in rats, and these steatotic allograft-induced liver inflammations and injuries were mediated by the activation of the caspase-11 non-canonical inflammasome and caspase-11 non-canonical inflammasome-induced IL-1β production [39]. TUDCA, an ER stress inhibitor, alleviated steatotic allograft injury and inflammation in rat livers and also inhibited the activation of the caspase-11 non-canonical inflammasome and IL-1β production [39], suggesting that ER stress and the activation of the caspase-11 non-canonical inflammasome in the liver play a critical role in steatotic allograft-induced liver inflammation and injury. Yin et al. reported that Jiangzhi Ligan Decoction (JZLGD), a Chinese herbal formula, affects NAFLD pathogenesis by regulating the caspase-11 non-canonical inflammasome in obese mice. JZLGD ameliorated NAFLD by reducing serum-lipid levels and lipid-droplet contents in the liver, resulting in the improvement of liver inflammation, injury, and function in HFD-fed rats [40]. Moreover, JZLGD inhibited the activation of the caspase-11 non-canonical inflammasome and, consequently, suppressed the proteolytic activation of GSDMD and the production of pro-inflammatory cytokines, IL-1β and IL-18, in the liver of HFD-fed NAFLD rats [40]. These results suggest that the caspase-11 non-canonical inflammasome is activated in NAFLD, leading to liver inflammation, injury, and dysfunction, and that the pharmacological effect of JZLGD on NAFLD is mediated by the inhibition of the caspase-11 non-canonical inflammasome in the liver.

Interestingly, an inhibitory role of the caspase-11 non-canonical inflammasome in NAFLD pathogenesis has also been reported. De Sant’Ana et al. demonstrated the protective effect of the caspase-11 non-canonical inflammasome in hepatic steatosis in obese mice. Lipid accumulation in the liver of standard-fat-diet (SFD)- and high-fat-diet (HFD)-fed *caspase-11^−/−^* mice [41]. Additionally, *caspase-11^−/−^* mice were more susceptible to HFD-induced obesity and exhibited enhanced development of hepatic steatosis in both SFD-fed and HFD-fed obese mice [41]. These results indicate that obesity and obesity-induced lipid accumulation and inflammation in the liver are associated with NAFLD development by regulating the function of the caspase-11 non-canonical inflammasome, which provides evidence of the crucial role of the caspase-11 non-canonical inflammasome in lipid accumulation in the liver and NAFLD pathogenesis. Drummer et al. performed genomic analyses and reported the regulatory role of the caspase-11 non-canonical inflammasome in the expression of genes upregulated in NAFLD using an NAFLD mouse model. Caspase-11 deficiency led to the upregulation and downregulation of genes associated with NAFLD-upregulated canonical and non-canonical inflammasomes, pro-inflammatory cytokines, and lipid peroxidation enzymes in mice [42]. These results indicate that the caspase-11 non-canonical inflammasome may play both aggravating and protective roles in NAFLD pathogenesis by modulating the expression of genes associated with NAFLD. Taken together, these studies suggest that the caspase-11 non-canonical inflammasome plays either pro- or anti-inflammatory roles in NAFLD, to exacerbate or protect against the disease.

### 3.2. NASH

NASH is an aggressive form of fatty liver disease characterized by liver inflammation and damage, which may progress to advanced scarring, known as cirrhosis; fibrosis; liver injury and failure; and hepatocellular carcinoma, eventually causing death [43,44]. NASH is an advanced and more serious form of NAFLD. Currently, a quarter of the world’s population has NAFLD, and approximately 20–25% of patients with NAFLD can develop NASH [45]. Epidemiological studies have revealed that more than 80% of patients with NASH suffer from obesity and hyperlipidemia, and approximately 50% of patients with NASH are also diagnosed with type-2 diabetes mellitus [46]. The major risk factors for the development of NASH include the increasing epidemics of obesity, dyslipidemia, and insulin resistance [47]. Therefore, the percentage of patients with NASH and their associated health-care costs will increase, thus warranting the early diagnosis and treatment of NASH.

Many studies have reported the roles of canonical inflammasomes in NASH [14,48,49,50]. Emerging studies have also demonstrated the regulatory role of non-canonical inflammasomes in NASH pathogenesis. Hendrikx et al. reported the role of the caspase-11 non-canonical inflammasome in hepatic inflammation and NASH development in mice lacking the low-density lipoprotein receptor (*Ldlr^−/−^*), which shows hepatic inflammation in Kupffer cells [51]. *Ldlr^−/−^* mice transplanted with *caspase-11^−/−^* bone marrow showed less-severe hepatic inflammation and NASH symptoms [52]. Cellular and molecular mechanism studies have revealed that Kupffer cells from *Ldlr^−/−^*/*caspase-11^−/−^* mice exerted less cholesterol accumulation and enhanced cholesterol efflux [52]. Moreover, bone marrow-derived macrophages (BMDMs) from *Ldlr^−/−^*/*caspase-11^−/−^* mice showed decreased autophagy induced upon oxidized low-density lipoprotein (oxLDL) stimulation [52]. These results suggest that the caspase-11 non-canonical inflammasome exacerbates NASH by increasing cholesterol crystal formation and decreasing cholesterol efflux, thereby inducing disturbed autophagy and inflammation in the liver. ER stress is an initiator of inflammatory signaling pathways and cell death and is linked to various diseases, such as obesity, type-2 diabetes mellitus, fatty liver diseases, and liver cancer [53,54]. Moreover, hepatic inflammation and cell death increase in NASH, eventually inducing liver injury and failure [55]. Lebeaupinn et al. investigated the functional crosstalk between ER stress and the caspase-11 non-canonical inflammasome in NASH pathogenesis in obese mice. LPS challenge induced liver inflammation and NASH-like pathological features by increasing ER stress and activating the caspase-11 non-canonical inflammasome, leading to subsequent hepatocyte pyroptosis and IL-1β secretion in obese mice [56]. ER stress inhibition by TUDCA decreased caspase-11 expression and caspase-11 non-canonical inflammasome activation, resulting in the amelioration of LPS-induced NASH-like pathological features in obese mice [56]. These results indicate that ER stress is a critical determinant of caspase-11 non-canonical inflammasome-activated hepatic inflammation and injury, leading to NASH pathogenesis. As described earlier, one of the most critical outcomes of caspase-11 non-canonical inflammasome activation is inflammatory cell death, known as pyroptosis [18,19]. Zhu et al. investigated the role of caspase-11 non-canonical inflammasome-mediated hepatic pyroptosis in NASH pathogenesis in methionine- and choline-deficient diet (MCD)-induced NASH in mice. The caspase-11 non-canonical inflammasome was activated in the liver of MCD-induced NASH mice; however, MCD-treated *caspase-11^−/−^* mice showed significantly reduced hepatic inflammation, pyroptosis, fibrosis, and injury [57]. Additionally, proteolytic activation of GSDMD and IL-1β secretion was markedly suppressed in MCD-treated *caspase-11^−/−^* mice, and overexpression of caspase-11 exacerbated MCD-induced hepatic steatosis in mice [57]. These results strongly indicate that the caspase-11 non-canonical inflammasome induces NASH by inducing hepatic inflammation and pyroptosis. Taken together, these studies suggest that the caspase-11 non-canonical inflammasome induces hepatic inflammation, injury, and NASH pathogenesis by promoting hepatic pyroptosis and the secretion of pro-inflammatory cytokines, as well as by orchestrating the differential expression of NASH-associated genes.

### 3.3. Inflammatory Liver Injury

Hepatic inflammation is considered a critical risk factor for liver injury and failure, which triggers various liver diseases associated with poor survival in patients [58,59]. Therefore, much effort has been made to understand the role and underlying mechanism of hepatic inflammation in inflammatory liver diseases and to develop effective therapeutics to treat inflammatory liver diseases. Canonical inflammasomes, especially the NLRP3 inflammasome, play key roles in hepatic inflammation, injury, and the pathogenesis of various inflammatory liver diseases [15,60,61,62]. However, the regulatory role of non-canonical inflammasomes in hepatic inflammation, injury, and inflammatory liver disease is poorly understood. Recent studies have demonstrated the protective and potential pharmacological effects of bioactive molecules on liver inflammation and injury. These bioactive molecules ameliorate inflammatory liver diseases by inhibiting the caspase-11 non-canonical inflammasome.

Heat shock protein A12A (HSPA12A) is a novel member of the HSP70 family that plays a role in the development of HFD-induced NAFLD and NASH [63]. Liu et al. investigated the protective function of HSPA12A in LPS-induced acute liver injury by inhibiting the caspase-11 non-canonical inflammasome in mice. *Hspa12a^−/−^* mice were more susceptible to LPS-induced acute liver inflammation and injury [64]. Activation of the caspase-11 non-canonical inflammasome was inhibited in the hepatocytes of *Hspa12a^−/−^* mice, resulting in the suppression of GSDMD pore formation and GSDMD pore-mediated hepatocyte pyroptosis [64]. These results suggest that HSPA12A plays a critical role against LPS-induced hepatic inflammation and liver injury by inhibiting activation of the caspase-11 non-canonical inflammasome and downstream inflammatory responses in hepatocytes.

Hepatic ischemia-reperfusion injury (IRI), a major complication of hepatic transplantation, resection, and hemorrhagic shock, often results in systemic hepatic inflammation and liver injury by activating macrophage-induced innate immune responses [65,66,67]. Lu et al. investigated the protective role of isoflurane, a halogenated anesthetic, and the mechanism underlying hepatic inflammation and IRI by targeting the caspase-11 non-canonical inflammasome in mice. Isoflurane alleviated hepatic IRI and liver injury in mice and decreased LPS-induced inflammation in hepatic macrophages [68]. Isoflurane also inhibited the activation of the caspase-11 non-canonical inflammasome, leading to the suppression of pyroptosis and the secretion of IL-1β and IL-18 in hepatic macrophages [68]. The above results indicate that isoflurane exerts a protective effect on hepatic IRI and liver injury by inhibiting caspase-11 non-canonical inflammasome-activated hepatic inflammation.

Samotolisib is a novel dual inhibitor targeting phosphoinositide 3-kinase (PI3K) and the mammalian target of rapamycin (mTOR), and has undergone several phase-II clinical trials as a potential treatment for different cancers. Zhao et al. screened out samotolisib after a systemic analysis of an FDA-approved compound library and reported the protective effect of samotolisib against LPS-induced hepatic inflammation and acute liver injury in mice. Samotolisib attenuated LPS-induced hepatic inflammation and acute liver injury, and improved survival in mice [69]. A mechanistic study revealed that samotolisib relieved the activation of the caspase-11 non-canonical inflammasome and hepatic pyroptosis by inhibiting PI3K/AKT/mTOR signaling pathways in the livers of LPS-injected mice [69], indicating that samotolisib protects against hepatic inflammation and acute liver injury by inhibiting the activation of the caspase-11 non-canonical inflammasome and hepatic pyroptosis-mediated liver injury. Taken together, these studies suggest that the caspase-11 non-canonical inflammasome promotes hepatic inflammation and liver injury in various inflammatory liver diseases and that agents targeting the caspase-11 non-canonical inflammasome may be potential therapeutics for inflammatory liver diseases.

The regulatory roles of the caspase-11 non-canonical inflammasome in the pathogenesis of NAFLD, NASH, and inflammatory hepatic injury are described in Figure 3.

## 4. Conclusions

Inflammasomes are inflammatory signalosomes that provide innate immunity against pathogens and cellular dangers, triggering a wide range of human diseases. Several studies have demonstrated that canonical inflammasomes, particularly NLRP3 inflammasomes, are key players in numerous inflammatory diseases [10,11], and sufficient evidence has demonstrated that canonical inflammasomes play critical roles in the pathogenesis of inflammatory liver diseases, such as NAFLD, NASH, and inflammatory liver injury [14,15,16]. The caspase-11 non-canonical inflammasome was recently discovered; therefore, its role is still largely unknown. Efforts have been made to demonstrate the role of the caspase-11 non-canonical inflammasome in inflammatory responses and diseases [19,70,71,72,73,74,75,76,77,78]. Interestingly, recent studies have also investigated the regulatory role of the caspase-11 non-canonical inflammasome in the pathogenesis of inflammatory liver diseases, which suggests that the caspase-11 non-canonical inflammasome is a key player in NAFLD, NASH, and liver diseases caused by inducing hepatic inflammation and GSDMD-dependent pyroptosis. In addition, hepatocytes expressing a low level of caspase-11 are resistant to pyroptotic cell death, and the overexpression of caspase-11 induces the activation of the caspase-11 non-canonical inflammasome in hepatocytes, resulting in GSDMD-dependent hepatocyte pyroptosis [79].

This review comprehensively summarizes and discusses the current knowledge of the regulatory role of the caspase-11 non-canonical inflammasome in the pathogenesis of inflammatory liver diseases and the underlying molecular mechanism (Table 1), which might improve our understanding of how the caspase-11 non-canonical inflammasome participates in exacerbating, or protecting from, inflammatory liver diseases. Despite the evidence from the studies discussed in this review, the regulatory roles of the caspase-11 non-canonical inflammasome in the pathogenesis of inflammatory liver diseases and the underlying mechanisms remain largely unknown. Moreover, the roles of caspase-4/5 non-canonical inflammasomes in human patients with inflammatory liver diseases have not yet been investigated. Therefore, further studies investigating the roles of the caspase-4/5/11 non-canonical inflammasome in various inflammatory liver diseases in appropriate animal models and human patients and the underlying mechanisms are required.

In conclusion, the caspase-11 non-canonical inflammasome is a key player in the pathogenesis of inflammatory liver diseases. The caspase-11 non-canonical inflammasome induces GSDMD-mediated pyroptosis and the secretion of pro-inflammatory cytokines, and caspase-11 non-canonical inflammasome-induced pyroptosis and pro-inflammatory cytokine secretion are independent of the canonical inflammasomes, which strongly suggests that the caspase-11 non-canonical inflammasome could be an independent therapeutic target of inflammatory liver diseases. Understanding the mechanisms that modulate the activity of the caspase-11 non-canonical inflammasome may contribute to the development of a wide range of therapeutic agents that can selectively target the caspase-11 non-canonical inflammasome in not only inflammatory liver diseases but also various inflammatory diseases caused by the activation of the caspase-11 non-canonical inflammasome.

## Figures and Tables

**Figure 1 ijms-23-04986-f001:**
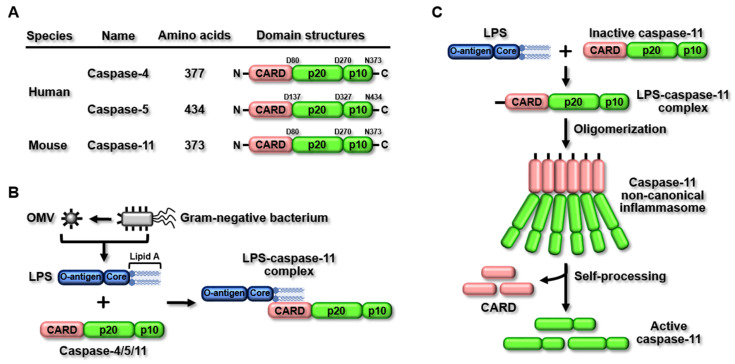
Structure and activation of the caspase-11 non-canonical inflammasome. (**A**) Human caspase-4, caspase-5, and mouse caspase-11 have similar domain structures, consisting of an N-terminal CARD, a p20 large catalytic domain, and a C-terminal p10 small catalytic domain. (**B**) Sensing LPS by caspase-11. Caspase-11 recognizes LPS by direct interaction between caspase-11 CARD with LPS lipid A. (**C**) Activation of the caspase-11 non-canonical inflammasome. Direct interaction between LPS and caspase-11 forms LPS–caspase-11 complexes, followed by oligomerization of LPS-caspase-11 complexes by CARD–CARD interaction. CARD domains are released from the LPS–caspase-11 oligomer through self processing, resulting in the production of active caspase-11.

**Figure 2 ijms-23-04986-f002:**
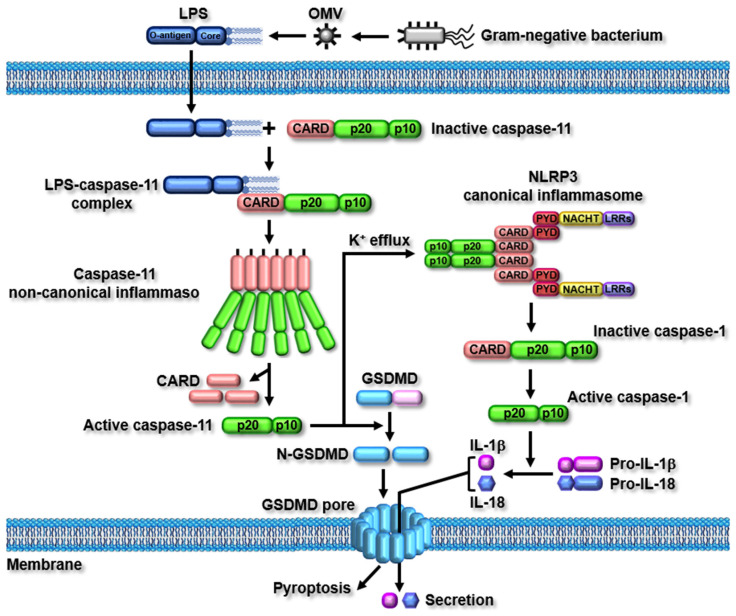
Caspase-11 non-canonical inflammasome-activated signaling pathways. Activation of caspase-11 non-canonical inflammasome induces the proteolytic processing of GSDMD. The processed N-GSDMD fragments move to the cell membrane and then generate GSDMD pores, leading to pyroptosis. Activation of caspase-11 non-canonical inflammasome also induces NLRP3 canonical inflammasome-mediated proteolytic activation of caspase-1, and the active caspase-1 induces the proteolytic maturation and secretion of IL-1β and IL-18 through GSDMD pores.

**Figure 3 ijms-23-04986-f003:**
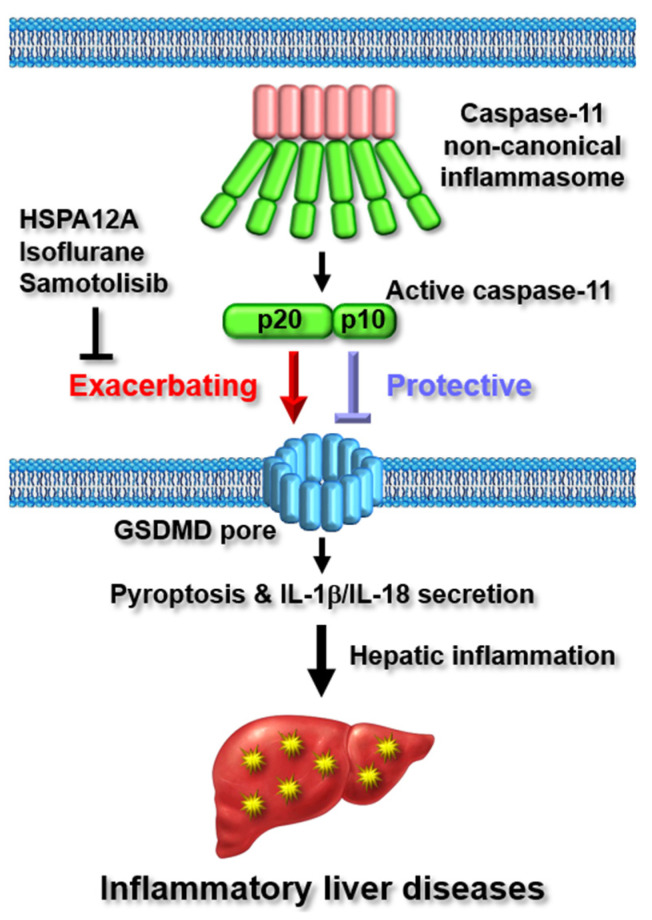
Graphical summary depicting the regulatory roles of the caspase-11 non-canonical inflammasome in inflammatory liver diseases.

**Table 1 ijms-23-04986-t001:** Regulatory roles of the caspase-11 non-canonical inflammasome in inflammatory liver diseases.

**Diseases**	**Roles**	**Models**	**Ref.**
NAFLD	Steatotic allograft increased ER stress, liver inflammation, and injury in rats TUDCA alleviated steatotic allograft injury and inflammation in rat livers TUDCA inhibited caspase-11 non-canonical inflammasome activation and IL-1β production	Steatotic liver transplanted rats	[39]
	JZLGD ameliorated NAFLD and improved liver function in HFD-fed rats JZLGD reduced the serum-lipid level and lipid-droplet contents in livers of HFD-fed rats JZLGD inhibited caspase-11 non-canonical inflammasome activation in HFD-fed NAFLD rats JZLGD inhibited proteolytic activation of GSDMD and production of IL-1β and IL-18 in HFD-fed NAFLD rats	HFD-fed obese rats	[40]
	Lipid accumulation in the livers of SFD- and HFD-fed caspase-11−/− mice Caspase-11−/− mice were more susceptible to HFD-induced obesity Caspase-11−/− mice exerted enhanced development of hepatic steatosis in both SFD-fed mice and HFD-fed obese mice	HFD-fed obese mice	[41]
	Gene expression of NAFLD-upregulated canonical and non-canonical inflammasomes, pro-inflammatory cytokines, and lipid peroxidation enzymes were upregulated and downregulated in caspase-11−/− NAFLD mice	HFD-fed obese and genetic-induced NAFLD mice	[42]
NASH	Ldlr−/−/caspase-11−/− mice showed less severe hepatic inflammation and NASH symptoms Kupffer cells from Ldlr−/−/caspase-11−/− mice exerted less cholesterol accumulation and enhanced cholesterol efflux BMDMs from Ldlr−/−/caspase-11−/− mice showed decreased autophagy	Ldlr−/− mice	[52]
	LPS induced liver inflammation and NASH-like pathological features in obese mice LPS increased ER stress and activated the caspase-11 non-canonical inflammasome, leading to hepatocyte pyroptosis and IL-1β secretion in obese mice ER stress inhibition by TUDCA decreased caspase-11 expression and caspase-11 non-canonical inflammasome activation in obese mice TUDCA treatment ameliorated LPS-induced NASH-like pathological features in obese mice	LPS-injected obese mice	[56]
	The caspase-11 non-canonical inflammasome was activated in livers of MCD-induced NASH mice MCD-treated caspase-11−/− mice showed reduced hepatic inflammation, pyroptosis, fibrosis, and injury Proteolytic activation of GSDMD and IL-1β secretion was suppressed in MCD-treated caspase-11−/− mice Overexpression of caspase-11 exacerbated MCD-induced hepatic steatosis in mice	MCD-treated mice	[57]
Inflammatory liver injury	Hspa12a−/− mice were more susceptible to LPS-induced acute liver inflammation and injury Activation of the caspase-11 non-canonical inflammasome was inhibited in the hepatocytes of the Hspa12a−/− mice GSDMD pore formation and hepatocyte pyroptosis was inhibited in the hepatocytes of the Hspa12a−/− mice	LPS-injected mice	[64]
	Isoflurane alleviated hepatic IRI and liver injury in mice Isoflurane decreased LPS-induced inflammation in hepatic macrophages Isoflurane inhibited caspase-11 non-canonical inflammasome activation in hepatic macrophages Isoflurane suppressed pyroptosis and secretion of IL-1β and IL-18 in hepatic macrophages	Hepatic IRI mice	[68]
	Samotolisib attenuated hepatic inflammation and acute liver injury in LPS-injected mice Samotolisib improved survival of LPS-injected mice Samotolisib relieved caspase-11 non-canonical inflammasome activation and hepatic pyroptosis by inhibiting PI3K/AKT/mTOR signaling pathways in livers of LPS-injected mice	LPS-injected mice	[69]

## Data Availability

Not applicable.

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
