# Peer review of "Regulatory Roles of Caspase-11 Non-Canonical Inflammasome in Inflammatory Liver Diseases"

_ijms, 2022, doi:10.3390/ijms23094986_

Round 1

Reviewer 1 Report

This manuscript of Y-S Yi is a well written and comprehensive review of the role of the caspase-11 non-canonical inflammasome in inflammatory liver disease. I have only a few suggestions:

  1. As stated, casp-11 is present in mice whereas 4, 5 are the corresponding caspases in humans. Obviously, despite the importance of basic science and use of mouse models the interest is on the latter, which is also supported by numerous references in this manuscript. Therefore, the author should consider to reflect this in the title, by e.g., “mouse c-11 and human c-4,5”, or something like “ what can we learn from murine c11” …
  2. An abbreviation for “caspase-11 non-canonical inflammasome” would be recommended.
  3. As pointed out, NAFLD and NASH are caused by high fat consumption, yet casps 11,4,5, are activated not by fat but by LPS which is a classical inflammatory mediator from gram-neg bacteria. The author should describe (better) why and how LPS would connect or exacerbate the inflammatory response caused by high fat, e.g., are you proposing that an additional bacterial infection is required for full-blown NAFLD, NASH?
  4. The potential therapeutics in chapter 3.3. should be placed into an own chapter.

Author Response

This manuscript of Y-S Yi is a well written and comprehensive review of the role of the caspase-11 non-canonical inflammasome in inflammatory liver disease. I have only a few suggestions:

1. As stated, casp-11 is present in mice whereas 4, 5 are the corresponding caspases in humans. Obviously, despite the importance of basic science and use of mouse models the interest is on the latter, which is also supported by numerous references in this manuscript. Therefore, the author should consider to reflect this in the title, by e.g., “mouse c-11 and human c-4,5”, or something like “ what can we learn from murine c11” …

- Author's Note: Thank you for your comment. Unfortunately, no study investigating the regulatory roles of human caspase-4/5 non-canonical inflammasomes in NAFLD, NASH, and inflammatory liver injury in human patients has been reported, so far, and all studies discussed in this article are about the roles of mouse caspase-11 non-canonical inflammasome in inflammatory liver diseases using the mouse models of NAFLD, NASH, and inflammatory liver injury. Although caspase-4/5 are human counterparts of the mouse caspase-11, it should be very careful to expand the experimental results of mouse caspase-11 to human caspase-4/5 without the experimental data in human patients. Therefore, the title was restricted only to mouse caspase-11 non-canonical inflammasome.

2. An abbreviation for “caspase-11 non-canonical inflammasome” would be recommended.

- Author's Note: I understand “caspase-11 non-canonical inflammasome” is quite long for readers. However, “caspase-11 non-canonical inflammasome” is widely used in this research field without using the abbreviation and is also understandable and acceptable for most readers. Therefore, your understanding and acceptance of using “caspase-11 non-canonical inflammasome” would be appreciated.

3. As pointed out, NAFLD and NASH are caused by high fat consumption, yet casps 11,4,5, are activated not by fat but by LPS which is a classical inflammatory mediator from gram-neg bacteria. The author should describe (better) why and how LPS would connect or exacerbate the inflammatory response caused by high fat, e.g., are you proposing that an additional bacterial infection is required for full-blown NAFLD, NASH?

- Author's Note: This is a very good point, and I appreciate your comment. Originally, mouse caspase-11 and human caspase-4/5 were discovered to be activated by direct interaction with LPS, an endotoxin of Gram-negative bacteria, and most of the previous studies have reported the roles of caspase-4/5/11 non-canonical inflammasomes in LPS-mediated inflammatory responses and diseases (mostly LPS-induced acute sepsis in a mouse model). Interestingly, recent emerging studies have reported the activation of the caspase-11 non-canonical inflammasome and its exacerbating roles in several inflammatory diseases, such as inflammatory gouty arthritis, diabetic nephropathy, multiple sclerosis, atherosclerosis, and inflammatory bowel disease that have nothing to do with LPS and Gram-negative bacterial infection (sterile inflammation), which strongly suggests that caspase-11 non-canonical inflammasome plays a critical role in not only non-sterile (LPS-induced) but also sterile (LPS-free) inflammation and inflammatory diseases. As you commented, the primary cause of NAFLD and NASH is high fat consumption and fat accumulation in the liver. However, it is well-known that high fat is a critical inducer of inflammatory responses, which might provide the connection between fat accumulation and sterile (LPS-free) inflammation in caspase-11 non-canonical inflammasome-mediated inflammatory liver diseases.

4. The potential therapeutics in chapter 3.3. should be placed into an own chapter.

- Author's Note: Thank you for your comment. The regulatory role of the caspase-11 non-canonical inflammasome in NAFLD and NASH was discussed in chapters 3.1 and 3.2, respectively. Chapter 3.3 also discussed the role of the caspase-11 non-canonical inflammasome in inflammatory liver injuries other than NAFLD and NASH, but all studies discussed in chapter 3.3 are about the therapeutic potentials of several molecules (heat shock protein A12A, isoflurane, samotolisib) on inflammatory liver injuries by targeting caspase-11 non-canonical inflammasome. Thus, chapter 3.3 consists of only the potential therapeutics for inflammatory liver injury, which cannot be separated into another chapter.

Reviewer 2 Report

This is a comprehensive, clearly written, up to date, tutorial review on regulatory roles of caspase 11 non-canonical inflammasome in inflammatory liver diseases. The figures and the Table are very helpful in understanding the ocean of data and mechanism.

Minor comments:

  1. Since it is an up to date review quite a recent publication should be added and discussed: Sun P et al. Hepatocytose are resistant to cell death from canonical and non-canonical inflamassome-activated pyroptosis. Cell Mol Gastroenterol Hepatol 2022;13:739757; https://doi.org/
    10.1016/j.jcmgh.2021.11.009
  2. No doubt the review is very clear and it is stressed that referring to the roles of non-canonical inflammasome there are mainly unknowns. Still based on what is known, even if speculative, the reader would like to get an impression whether non canonical inflammasome can stand alone and thus be an independent target of therapy in the relevant liver pathology. 

Author Response

This is a comprehensive, clearly written, up to date, tutorial review on regulatory roles of caspase 11 non-canonical inflammasome in inflammatory liver diseases. The figures and the Table are very helpful in understanding the ocean of data and mechanism.

Minor comments:

1. Since it is an up to date review quite a recent publication should be added and discussed: Sun P et al. Hepatocytose are resistant to cell death from canonical and non-canonical inflamassome-activated pyroptosis. Cell Mol Gastroenterol Hepatol 2022;13:739–757; https://doi.org/10.1016/j.jcmgh.2021.11.009

- Author's Note: Thank you for your comment. The literature you suggested has been added to the manuscript (Ref. 78).

2. No doubt the review is very clear and it is stressed that referring to the roles of non-canonical inflammasome there are mainly unknowns. Still based on what is known, even if speculative, the reader would like to get an impression whether non canonical inflammasome can stand alone and thus be an independent target of therapy in the relevant liver pathology.

- Author's Note: This is a very good point, and I appreciate your comment. As discussed in the manuscript, the caspase-11 non-canonical inflammasome induces the GSDMD-mediated pyroptosis and the secretion of pro-inflammatory cytokines, and the caspase-11 non-canonical inflammasome-induced pyroptosis and pro-inflammatory cytokine secretion are independent of the canonical inflammasomes, which strongly suggests that caspase-11 non-canonical inflammasome could be an independent therapeutic target of inflammatory liver diseases. This has been discussed in the Conclusion section.